# Hsp70 Peptides Induce TREM-1-Dependent and TREM-1-Independent Activation of Cytotoxic Lymphocytes

**DOI:** 10.3390/ijms26199750

**Published:** 2025-10-07

**Authors:** Daria M. Yurkina, Elena A. Romanova, Aleksandr S. Chernov, Irina S. Gogleva, Anna V. Tvorogova, Alexey V. Feoktistov, Rustam H. Ziganshin, Denis V. Yashin, Lidia P. Sashchenko

**Affiliations:** 1Institute of Gene Biology (RAS), Moscow 119334, Russia; yrkina121@gmail.com (D.M.Y.); elrom4@rambler.ru (E.A.R.); annatvor@mail.ru (A.V.T.); a.feo95@mail.ru (A.V.F.); sashchenko@genebiology.ru (L.P.S.); 2Shemyakin & Ovchinnikov Institute of Bioorganic, Chemistry, Russian Academy of Sciences, ul. Miklukho-Maklaya, 16/10, Moscow 117997, Russia; alexandrchernov1984@gmail.com (A.S.C.); gogleva.irina2017@yandex.ru (I.S.G.); ziganshin@mail.ru (R.H.Z.)

**Keywords:** TREM-1, Hsp70, TKD peptide, cytotoxicity, tumor cells, apoptosis, necroptosis, short peptides

## Abstract

The novel data show that the Hsp70 protein is a potent activator of the immune system. Using limited trypsinolisis, we have identified the epitopes of Hsp70 responsible for TREM-1-dependent and TREM-1-independent cytotoxicity. The 11aa N9 peptide (AMTKDNNLLGR) contains nine amino acids that correspond to the amino acid sequence of the known TKD peptide. Also, like TKD, this peptide does not interact with the TREM-1 receptor but activates CD94+ NK cells that kill tumor cells by secreting granzymes and inducing apoptosis. The 16aa peptide N7 (SDNQPGVLIQVYEGEK) interacts with the TREM-1 receptor and induces the activation of NK cells and cytotoxic T lymphocytes at different time points. T-lymphocytes activated by this peptide induce two alternative processes of cell death in HLA-negative tumor cells, apoptosis and necroptosis, through the interaction of the FasL lymphocyte with the Fas receptor of the tumor cell. A shortened fragment of this peptide, N7.1 (SDNQPGVL), has been identified that inhibits the interaction of TREM-1 with its ligands. This peptide has shown protective effects in the development of sepsis in mice. The results obtained can be used in antitumor and anti-inflammation therapy.

## 1. Introduction

The study of the interaction of pro-inflammatory receptors with their ligands attracts wide attention in modern immunology. The binding of the ligand to the exodomain of the receptor can be considered as the “first point” of initiation of the regulatory signal. It is the upstream point at which the intracellular signal induction can be activated or inhibited. For a detailed study of the mechanisms of receptor activation under the action of ligands, it is important to expand the spectra of ligands and identify epitopes responsible for the functional activity of these ligands. Peptides corresponding to the amino acid sequence of these epitopes can be used to create drugs that regulate the processes induced by these receptors. To search for new ligands, it seems promising to search for new activities of multifunctional proteins. A striking example of such a protein is the main heat shock protein Hsp70.

Hsp70 belongs to the extensive family of chaperone proteins [1]. Its functional activity varies depending on the localization of this protein in the cell. Cytoplasmic, membrane-bound, and soluble Hsp70 are distinguished [2,3]. Cytoplasmic Hsp70 is aimed at maintaining the normal functioning of cells, including tumor cells. This protein is responsible for proper protein folding during normal and stressful conditions [4]. It can also inhibit the development of apoptosis during cell death [5].

Hsp70 lacks a transmembrane sequence, but is highly expressed on the surface of tumor cells. Multhoff et al. believe that Hsp70 is transported to plasma membranes by binding to other proteins [6]. The active transport of Hsp70 to the membrane leads to its release into the extracellular environment. Membrane and soluble Hsp70 are involved in the regulation of innate and acquired immunity. The interaction of Hsp70 with antigen-presenting cells leads to the stimulation of pro-inflammatory cytokines [7]. The ability of Hsp70 to bind antigenic peptides in the peptide-binding domain allows this protein to participate in the presentation of peptides and activate regulatory and effector lymphocytes in adaptive immunity [8]. The interaction of Hsp70 with the CD94 lectin receptor increases the binding strength of the CD94/NKG2D and NKG2C heterodimeric receptors on the surface of NK cells, activating their cytotoxic effect. The interaction of granzyme B secreted by NK cells with membrane Hsp70 promotes the development of apoptosis in tumor cells [9,10]. It was shown that an ex vivo stimulation of patient-derived NK cells with interleukin (IL)-2 and Hsp70-derived peptide TKD (TKDNNLLGRFELSG, aa450–461) results in a significant upregulation of activating receptors, including CD94 and CD69, which triggers exhausted NK cells to target and kill malignant solid tumors expressing membrane Hsp70 (mHsp70). Activation of NK cells via TKD leads to enhanced production and release of lytic granules and pro-inflammatory cytokines against tumor cells [10]. Membrane Hsp70 acts as a tumor-specific antigen of NK cells and CD4+CD25+ lymphocytes. Membrane Hsp70 is widely used for tumor detection in antitumor therapy [11,12,13]. Hsp70 receptors on immune cells are described. These include CD14, Toll-like receptors (TLR2 and TLR4), CD91, CD40, and RAGE [14,15,16,17,18]. It has recently been shown that Hsp70 is a ligand of the TREM-1 innate immunity receptor [19].

TREM-1 was first described in 2000 as a key regulator of inflammation [20]. TREM-1 is mainly a receptor for innate immune cells. It is present on the surface of monocytes, neutrophils, dendritic cells, and NK cells and, to a lesser extent, on some subpopulations of T and B lymphocytes [21]. Although TREM-1 is usually considered a pro-inflammatory receptor, it has recently been shown that it can exhibit bifunctional activity. It not only promotes the release of inflammatory cytokines but also induces the appearance of subpopulations of cytotoxic lymphocytes that kill HLA-negative tumor cells [22]. Activation of TREM-1 begins with dimerization of the receptor, followed by dissociation of the soluble exodomain. Further activation of the cascade of tyrosine protein kinases leads to the appearance of the transcription factor NFkB [23]. NFkB induces the expression of pro-inflammatory cytokine genes, which causes the development of inflammation. In synergy with TLR4, TREM-1 is responsible for the overexpression of proinflammatory cytokines (cytokine storm) [24]. This phenomenon can lead to irreversible consequences for the organism. Overexpression of TREM-1 has been associated with sepsis, arthritis, and viral infections, including COVID-19 and EBOLA [25,26,27]. Dissociated from the cell surface, sTREM-1 is considered a marker of severe viral diseases [28,29]. The release of cytokines can also lead to the activation of regulatory lymphocytes that induce effector cytotoxic lymphocytes against tumor cells in a TCR-MHC independent manner. HLA-negative tumor cells are out of immune control; they are not recognized by classical cytotoxic lymphocytes and play a significant role in tumor progression [22].

TREM-1 ligands have been described, including actin, RNA-binding protein (eCIRP), multifunctional nuclear protein HMGB1, and innate immunity protein PGLYRP1 [30,31,32,33]. It has also been shown that the surface glycoproteins of Ebola and Marburg viruses react with TREM-1 [34]. The peptide LP17 was described as the TREM-1 signaling inhibitor [24]. Recently, it was shown that the Hsp70 protein, when interacting with peripheral blood mononuclear cells (PBMC), activates TREM-1-dependent and TREM-1-independent cytotoxic lymphocytes that kill HLA-negative tumor cells [19].

The aim of this work was to identify the Hsp70 epitopes responsible for the functional activity of these lymphocytes. The main tasks were as follows: (1) establishment of the amino acid sequence of these epitopes; (2) characterization of their interaction with TREM-1; (3) description of subpopulations of cytotoxic lymphocytes and cytotoxic processes in tumor cells induced by the action of these peptides.

## 2. Results

### 2.1. Hsp70 Peptides Obtained by Trypsin Hydrolysis Induce TREM-1-Dependent and TREM-1-Independent Activation of Cytotoxic Lymphocytes

First, we identified Hsp70 epitopes involved in the activation of cytotoxic lymphocytes. Limited trypsinolysis of full-size Hsp70 was used for this purpose. The recombinant human Hsp70 was digested by trypsin protease according to [35], and then the resulting mixture of peptides was separated using gel filtration on a Superdex-Peptide column. Aliquots from each fraction were added to PBMC, and after 6 days of incubation, the cytotoxic activity of these cells was determined. For this purpose, PBMC activated by these peptides was added to K562 HLA-negative tumor cells, and cytotoxicity was measured. From the results shown in Figure 1, it can be seen that the peptides contained in fractions No. 7 and No. 9 induce cytotoxic activity of lymphocytes. A more detailed cytotoxicity study suggested that the identified peptides interact with various receptors and activate various subpopulations of lymphocytes. These lymphocytes induce various mechanisms of programmed cell death in tumor cells.

Figure 2 shows the cytotoxic activity of PBMC activated by peptides of fractions No. 9 and No. 7 for 6 days. It can be seen that the cytotoxicity of lymphocytes under the action of peptides of fraction No. 9 reaches its maximum value after 3 h of their interaction with target tumor cells. The TREM-1 receptor inhibitor LP17 peptide and antibodies to FasL have no effect on the cytotoxic effect of these lymphocytes. However, the removal of activated PBMC CD94+ lymphocytes from the population, as well as the addition of antibodies to granzyme B, completely blocks cytotoxicity (Figure 2a).

The action of peptides of fraction No. 7 has the opposite effect on the cytotoxicity of PBMC. (Figure 2b). It can be seen that activation of this fraction of peptides induces two cytotoxic processes occurring at different rates. The maximum activity is observed after 3 h and 20 h of interaction with PBMC. The cytotoxic activity of lymphocytes disappears in the presence of LP17 and antibodies to FasL. Removal of CD94+ lymphocytes from the PBMC population, as well as the addition of antibodies to granzyme B, does not lead to a sharp decrease in cytotoxic activity.

These results indicate that different peptides activate different subpopulations of cytotoxic lymphocytes that kill the same tumor cells. The K562 erythroblastoma cell line is a target for both NK cells and non-canonical CD4+ CD8+ T lymphocytes.

Using MALDI-TOF analysis, the amino acid sequence of the peptides contained in these fractions was determined. The fraction №7 containing the 16-membered peptide (SDNQPGVLIQVYEGEK, designated N7) is located on the peptide-binding domain of Hsp70. This peptide is unique; its effect on the cytotoxicity of lymphocytes has not been described. The structure of the peptide from the No. 9 fraction corresponded significantly to the structure of the well-studied TKD. Eight of the 11 amino acids of the N9 peptide overlap with the amino acid sequence of the TKD peptide.

Next, we investigated the mechanisms of cytotoxic action of lymphocytes activated by TKD and synthesized peptide N7.

### 2.2. Activated TKD Lymphocytes Cause Apoptosis in Target Tumor Cells

TKD is known to be responsible for the interaction of Hsp70 with NK cells and their activation. Therefore, we first investigated the dependence of NK cell activation on incubation time with TKD. For this purpose, PBMC was incubated with TKD, isolated NK cells were added to K562 target cells, and cytotoxic activity was determined after 3 h of incubation. It can be seen that the activity of NK cells increases in proportion to the increase in incubation time. The maximum cytotoxicity is achieved after 5 days of incubation of NK cells with TKD and persists for 6 days (Figure 3a).

Next, we investigated the mechanisms of cytotoxic action induced by TKD-activated NK cells. The results are shown in Figure 3b,c. It can be seen that the TREM-1 receptor inhibitor, the LP17 peptide, does not inhibit the cytotoxic effect of TKD-activated NK cells. However, incubation of activated TKD PBMC with CD94 antibodies leads to the disappearance of cytotoxic activity. These data suggest that the CD94 antigen is a receptor for TKD, and TKD does not interact with TREM-1, although full-length Hsp70 is a ligand of this receptor. The induction of target cell death disappears in the presence of antibodies to granzyme B, which suggests that, in this case, the cytotoxicity of NK cells is due to the secretion of granzyme B into the immunosynapse zone.

The results shown in Figure 3c indicate that TKD-activated NK cells induce caspase-dependent apoptosis. The caspase 3 inhibitor completely blocks cell death. Pre-incubation of K562 cells with MicA antibodies and NK cells with antibodies to NKG2D and CD94 antigens also completely blocks cytotoxicity. These results show that the interaction of both subunits of the CD94/NKG2D heterodimeric receptor with the non-canonical HLA MicA molecule is essential for the induction of a cytotoxic signal.

Thus, it was demonstrated that TKD is not an Hsp70 epitope responsible for the interaction of this protein with TREM-1. The interaction of TKD with the CD94/NKG2D heterodimeric receptor subunit, CD94, is essential for both NK activation and cytotoxic signal induction in tumor cells.

Next, we investigated the interaction of the N7 peptide with the TREM-1 receptor.

### 2.3. Peptide N7 Interacts with TREM-1 in Solution and on the Cell Surface

First, we studied the affinity of the N7 peptide to the TREM-1 receptor. For this purpose, microscale thermophoresis was used. Figure 4a shows the signals of binding of peptide N7 to the receptor. A clear dependence of the thermodynamic signal on the concentration of the peptide can be seen. The KD was 1.6 nM, which indicates the high affinity and stability of the complex with the receptor.

Figure 4b shows the interaction of TREM-1 with TKD. You can also see a clear thermodynamic signal depending on the concentration of the peptide. However, the KD was 3 mkM, which is 1000 times higher than the KD for the N7 peptide. These data indicate the low affinity of this peptide to the receptor and the impossibility of forming a stable complex.

The binding of the peptide to cells was studied using confocal microscopy (Figure 5). It can be seen that the TREM-1 receptor is present on the cells (green), the N7 peptide binds on the cell surface (red). The colocation of the peptide with the receptor (yellow) indicates their interaction. Statistical analysis has shown that 62% of all observed cells were double-stained (*n* = 4) (Appendix A).

Incubation of the N7 peptide with cells in the presence of the BS^3^ crosslinking agent, followed by analysis of the N7-TREM-1 complex, made it possible to determine the molecular mass of this complex. After performing PAAG and Western blot with specific antibodies, it can be seen that the molecular weight of the complex is 52 kDa, which corresponds to its dimer. It can be assumed that after binding to the N7 peptide, TREM-1 dimerizes (Figure 6, Appendix A).

As mentioned above, the TREM-1 receptor dimerizes after binding to the ligand. Further signal transmission is accompanied by activation of the MMP9 membrane protease, which causes dissociation of the TREM-1 exocellular domain. The activation of this receptor is judged by the appearance of soluble sTREM-1. Figure 7 shows ELISA data indicating the appearance of sTREM-1 in a conditioned medium of monocytes incubated with the N7 peptide. The yield of sTREM-1 is dose-dependent and depends on the concentration of the N7 peptide.

Thus, it was shown that the N7 peptide is the Hsp70 epitope responsible for the interaction of this ligand with TREM-1. It interacts with the receptor and induces its activity.

Next, the effect of this peptide–receptor interaction on the activation of cytotoxic lymphocytes was investigated.

### 2.4. Peptide N7 Induces TREM-1-Dependent Cytotoxicity of Lymphocytes

First, the ability of synthetic peptide N7 to induce TREM-1-dependent cytotoxicity was investigated. It can be seen that after a 6–day incubation of PBMC with this peptide, cytotoxic lymphocytes appear in the population, killing MHC-negative K562 tumor cells. The cytotoxic signal is induced through the TREM-1 receptor. It can be seen that the inhibitory peptide LP17 and antibodies against this receptor completely block the cytotoxic effect (Figure 8a). As a control, PBMC were incubated with the N7 peptide in the presence of a TLR receptor inhibitor. It can be seen that the cytotoxic activity of lymphocytes does not change in the presence of this inhibitor; that is, TLR is not involved in the induction of cytotoxicity.

As was shown for the full-length Hsp70, the N7 peptide activates different subpopulations of lymphocytes at different time intervals (Figure 8b, Appendix A). On the 4th day, NK cells reached their maximum activity, and cytotoxic CD4+-T lymphocytes were also activated. No traces of cytotoxic activity of CD8+-T lymphocytes were found.

On the 6th day of incubation, the cytotoxic activity of NK cells completely disappeared, the activity of CD4+-T-lymphocytes remained, and the cytotoxicity of CD8+-T-lymphocytes appeared.

A more detailed analysis of the cytotoxic effect showed that the activity of NK cells is blocked by antibodies to granzyme B, the NKG2D receptor and the non-canonical MHC molecule MicA protein, as well as an inhibitor to caspase 3. (Figure 8c). In our experiment, as previously described, NK cells also use the NKG2D receptor to recognize MicA on tumor cells, followed by secretion granzyme B into the immunosynapse zone and induction of caspase-dependent apoptosis in tumor cells.

The NKG2D receptor is present on the membrane of CD8+T lymphocytes as well as on the membrane of NK cells, and its interaction with MicA is also essential for the induction of cytolysis. However, in this case, this interaction does not lead to the secretion of granzymes but to the activation of FasL on the surface of lymphocytes. The activation of FasL on the surface of CD4+-T lymphocytes requires the interaction of membrane Tag7 (PGLYRP1) with the Hsp70 molecule present on the surface of target cells of the K562 lineage. In both cases, activated FasL interacts with the Fas receptor on the surface of the tumor cell, inducing apoptosis after 3 h of interaction with tumor cells and necroptosis after 20 h. It can be seen that antibodies to NKG2D, Tag7, Hsp70, FasL, MicA, as well as an inhibitor of caspase 3 and an inhibitor of RIP1 kinase, prevent the induction of cytotoxic signal.

Thus, it can be assumed that the N7 peptide is an epitope of Hsp70 responsible for the activation of the TREM-1 receptor and, like full-length Hsp70, is able to activate the cytotoxic activity of NK cells and CD8+ and CD4+T lymphocytes against MHC-negative tumor cells.

### 2.5. The Shortened Fragment of the N7 Peptide Blocks the Binding of TREM-1 to Ligands and Has a Protective Effect in the Development of Sepsis

Further, structural fragments of the N7 peptide responsible for binding to TREM-1 were identified. For this purpose, it was divided into two parts and two peptides with N-terminal and C-terminal amino acid sequences were synthesized. The ability to activate the cytotoxic activity of three TREM-1 ligands: peptide N7, nuclear protein HMGB1, and full-length Hsp70 was investigated. In the presence of these peptide fragments, ligands were added to PBMC, previously pre-incubated with shortened peptides, and after 6 days of incubation, the cytotoxic activity of lymphocytes was determined (Figure 9).

It can be seen that the N-terminal fragment of N7 peptide N7.1 (SDNQPGVL) blocks the activation of lymphocytes by all the ligands studied. Thus, it can be assumed that the shortened peptide N7.1 interacts with the TREM-1 region responsible for binding to ligands and prevents their interaction. 

### 2.6. Investigation of the Anti-Inflammatory Effect of Peptide N7.1 on a Model of LPS-Induced Sepsis

Septic shock develops when bacterial endotoxin lipopolysaccharide (LPS), a component of the cell wall of gram-negative bacteria, enters the body. The destructive role of LPS is to stimulate the massive production of pro-inflammatory cytokines, primarily TNF.

After administration of *S. enterica* LPS to animals, an examination was performed at 2, 8, 12, 24, 48, 72, and 96 h and the number of dead animals estimated. The results are shown in Figure 10.

Intact animals, without induction of sepsis, demonstrated normal behavior throughout the experiment: high motor activity, smooth coat, clean mucous membranes of the eyes and nose, normal intake of feed and water, and adequate response to external stimuli (capture of the animal by the tail with tweezers).

During the first 2 to 8 h of observation, all animals treated with LPS showed a decrease in motor activity; the animals huddled into groups and their coats acquired an untidy, disheveled appearance. After 24 h, purulent discharge on the mucous membranes of the eyes was recorded in animals from all groups, and the animals were inactive.

In the saline group, death began on day 2 after sepsis induction, and by day 5, only two mice (29%) had survived. Administration of the Hsp70 N7 peptide to mice with induced sepsis had a protective effect: only one mouse died on day 3 after LPS administration, and the percentage of surviving animals was 86%.

The Hsp70 peptide demonstrated high activity in the LPS-induced sepsis model in mice and contributed to the inhibition of septic shock in experimental animals.

## 3. Discussion

The main achievement of this work is the identification of two peptide fragments that initiate the cytotoxic effect of lymphocytes and interact with various receptors. The 11-membered peptide N9 (AMTKDNNLLGR) contains 8 amino acids corresponding to the amino acid sequence of the well-studied peptide TKD. The N9 peptide can be considered as a shortened TKD peptide. Both the N9 and TKD peptide interact with the CD94 receptor and have the same effect on the activity of NK cells; therefore, it can be assumed that an eight-membered shortened peptide present in the structure of both TKD and N9 is responsible for this activity.

TKD was first described by Multhoff and is involved in carrying out a variety of Hsp70 functions. The activation of cytotoxic processes that cause cell death is particularly interesting. Thus, TKD is responsible for the activation of NK cells through the CD94 receptor, the passage of granzyme B through the cell membrane, and the formation of the Tag7-Hsp70 cytotoxic complex. We confirmed that TKD interacts with the CD94 receptor and showed that TKD, when incubated with human PBMC ex vivo, induces stable activity of NK cells, but not cytotoxic T lymphocytes. It was also shown that the CD94 receptor is essential not only for the activation of NK cells, but also for the transmission of a cytotoxic signal during the interaction of already activated TKD NK cells with tumor cells.

The N7 peptide does not interact with either CD94+ or TLR receptors. It specifically binds to TREM-1 and, unlike TKD, induces the activation of a fairly wide range of cytotoxic lymphocytes: classical NK cells and non-canonical CD8+ and CD4+ lymphocytes, which kill MHC-negative cells that have escaped immune control. Like full-length Hsp70 and other ligands (Tag7, N3, HMGB1), the N7 peptide has a high affinity for TREM-1, binds to it on the cell surface, forming a dimer and causes sTREM-1 dissociation from the cytoplasmic membrane of the macrophage. As has been shown for other ligands, lymphocytes activated by this peptide induce alternative cell death processes in tumor cells: caspase-dependent apoptosis and RIP1K-dependent necroptosis.

Thus, the N7 peptide can be considered a new activator of cytotoxic lymphocytes against MHC-negative tumor cells and used in the development of drugs for antitumor therapy.

A shortened fragment of the N7 peptide (SDNQPGVL) was identified and synthesized, which binds to the active site of the receptor and competes with other ligands. This peptide had a protective effect in the development of sepsis. As mentioned above, the synergism of the pro-inflammatory TLRs and TREM-1 receptors has been described. Activation of TLRs leads to an increase in the amount of TREM-1 on the cell surface of macrophages, for which proteins of dead cells can be ligands. In this case, simultaneous activation of pro-inflammatory receptors leads to increased secretion of inflammatory cytokines (cytokine storm) and is often fatal. We activated TLR receptors by adding LPS to the animal and detected the development of fatal inflammation. The survival rate of these mice was 29%. The protection of the TREM-1 receptor by adding a shortened peptide fragment appears to block the loop of inflammatory process enhancement. In this case, the survival rate was 86%. This peptide can be used to create anti-inflammatory drugs.

## 4. Materials and Methods

### 4.1. Cell Culture and Sorting

Human leukemic cell line K562 and THP-1 cell line were cultured in RPMI-1640 (Himedia Laboratories Private Limited, Maharashtra, India) and 10% FCS (Cytiva Livescience™, Marlborough, MA, USA). This cell line was obtained from the cell line collection of N. N. Blokhin National Medical Research Center of Oncology of the Ministry of Health of Russia. Human peripheral blood mononuclear cells (PBMC) were isolated from the total leukocyte pool of healthy donors by centrifugation in a Ficoll-Paque density gradient (Cytiva Livescience™, Marlborough, MA, USA), and cultured at a density of 4 × 10^6^ cells/mL in RPMI-1640 (Himedia Laboratories Private Limited, Maharashtra, India) with 10% FCS (Cytiva Livescience™, Marlborough, MA, USA) with peptide fractions of Hsp70, Hsp70, HMGB1, TKD peptide, synthetic Hsp70 peptides in a concentration of 10^−9^ M (for all agents) for 6 days. The cells were sorted using sets of magnetic beads for isolation of CD14+, CD16+, CD56+, CD8+, and CD4+ according to the manufacturer’s protocol (all were produced by Thermo Fisher Scientific, Waltham, MA, USA).

All procedures performed were conducted in accordance with the Declaration of Helsinki and approved by the Ethics Committee of the Institute of Gene Biology (Moscow), protocol no. 23 dated 15 August 2024.

### 4.2. Proteins and Antibodies

The cDNAs for the recombinant human Hsp70 were subcloned into pQE-31 and expressed in *Escherichia coli* M15 (pREP4) (Qiagen, Germantown, MD, USA). Hsp70 was produced as described [36]. HMGB1 was produced as described [30]. sTREM-1 was obtained according to [22]. The bacterial LPS was not detected in the recombinant Hsp70 preparation by Pierce Chromogenic Endotoxin Quant Kit (Thermo Fisher Scientific, Waltham, MA, USA). Monoclonal antibodies to TREM-1 and polyclonal antibodies to Hsp70 were obtained from Thermo Fisher Scientific (Thermo Fisher Scientific, Waltham, MA, USA).

### 4.3. Peptides

Protein Hsp70 was hydrolyzed at 37 °C for 3.5 h at a 1:10 trypsin/protein ratio (*w*/*w*) in 50 mM (NH3)HCO3 (pH 8.0). The hydrolysate was then separated on a Superdex-Peptide column. Peptides 7 and 9 were produced synthetically via an automated peptide synthesizer, adhering to the Fmoc methodology, with HATU serving as the coupling reagent. The activated resin underwent the attachment of C-terminal amino acids with DIPEA for a duration of 2 h. After synthesis, the protected peptidyl polymer was washed with diethyl ether, dried, and treated with a mixture of TFA/DTT/H2O/TIS (150/4/3/0.5 wt. %) (15 mL of the mixture per g of peptidyl polymer) for 2 h. The solution was filtered; the untreated peptide was precipitated with a tenfold volume of diethyl ether and kept at a temperature of 4 °C for 8 h. The precipitated peptide was centrifuged, washed three times with diethyl ether, and dried under vacuum. The untreated peptide was purified by HPLC in a gradient of 5–55% acetonitrile and lyophilized. TKD peptide was from Thermo Fisher Scientific (Boston, MA, USA).

### 4.4. Affinity Chromatography, Immunoadsorption, and Immunoblotting

Monocytes were incubated with the N7 peptide (100 nM) in the presence of BS^3^ (Thermo Fisher Scientific, Boston, MA, USA), lysed in RIPA buffer (Sigma-Aldrich, St. Louis, MO, USA) and purified using Dynabeads (M-280 sheep anti-rabbit IgG; Dynal Biotech ASA, Oslo, Norway), conjugated with anti-TREM-1 antibody in accordance with the manufacturer’s protocol. This material was resolved into 12% SDS-PAGE, followed by Western blotting. To detect the resulting complex N7-TREM1, primary mouse anti-TREM-1 antibodies (1:1000, overnight) followed by secondary HRP-conjugated anti-mouse antibody (GE Healthcare, Chicago, IL, USA; 1:10,000; 1 h) were used for one fraction, and primary rabbit anti-Hsp70 antibodies (1:1000, overnight) followed by secondary HRP-conjugated anti-rabbit antibody (GE Healthcare, Chicago, IL, USA; 1:10,000; 1 h) were used for another fraction. The results were visualized using the ECL Plus kit (GE Healthcare, Chicago, IL, USA) in accordance with the manufacturer’s protocol. Chemiluminescence was recorded using iBright (Thermo Fisher Scientific, Boston, MA, USA).

### 4.5. Cytotoxicity Assays

For cytotoxicity tests, cells of the K562 line were cultured in 96-well plates (Guangzhou Jet Bio-Filtration Co., Guangzhou, China) (6 × 10^4^ per well) mixed with lymphocytes added at a 20:1 ratio and incubated at 37 °C in a 5% CO_2_ atmosphere for 3 to 24 h). In the inhibition assays, cells were pre-incubated for 1 h with the LP17 (LQVTDSGLYRCVIYHPP) (Thermo Fisher Scientific, Boston, MA, USA), abTREM1 (ABclonal, Woburn, MA, USA), anti-NKG2D, anti-granzyme B, caspase 3 inhibitor Ac-IEID-CHO, RIP1 kinase inhibitor necrostatin1, TLR4 inhibitor, anti-MicA, anti-FasL, anti-CD94 antibody (concentration was 20 μg/mL for all inhibitors and all from Thermo Fisher Scientific, Waltham, MA, USA) and then the lymphocytes were added.

Cytotoxicity was measured using a Cytotox 96 analysis kit (Promega, Madison, WI, USA) after 24 h of incubation in accordance with the manufacturer’s protocol [37].

### 4.6. Microscale Thermophoresis

The purified sTREM-1 protein was labeled using an Alexa FluorTM 488 Protein Labeling Kit (Life Technologies Corporation, Eugene, OR, USA) in accordance with the manufacturer’s protocol.

The N7 peptide and TKD (C = 200 nM for all) were incubated for 30 min in the dark at room temperature in 16 different concentrations obtained through sequential dilution, starting with the highest soluble concentration. The samples were transferred to glass capillaries (Monolith NT Capillaries, NanoTemper Technologies GmbH, München, Germany) and then analyzed using a Nano-Temperature Monolith NT 115 device (8% IR laser power). The signal quality was monitored using a NanoTemper Monolith device to detect possible autofluorescence of the ligand, aggregation, or changes in the photobleaching rate. The experiments were carried out in at least three replicates and processed using affinity analysis software (MO Control v.1.6.1, NanoTemper Technologies GmbH, München, Germany) [38].

### 4.7. Confocal Microscopy

The THP-1 cells were grown on glass coverslips, and peptide N7 was added. After 1 h of incubation, the cells were washed and fixed with 4% formaldehyde for 15 min, then the monocytes were rinsed three times with PBS, and the samples were placed into blocking solution (1% BSA in PBS) for 30 min at room temperature. The TREM-1 receptor was stained with monoclonal mouse antibodies against TREM-1 and a Goat anti-mouse IgG (H+L) Cross-Adsorbed Secondary Antibody, Alexa FluorTM 488 (Molecular Probes By Life Technologies, Carlsbad, CA, USA). The peptide N7 was stained with polyclonal rabbit antibodies against Hsp70 and a Goat anti-Rabbit IgG (H+L) Cross-Adsorbed Secondary Antibody, Alexa FluorTM 633. Following washing with PBS, the coverslips were embedded in ProLong Gold (Thermo Fisher Scientific, Waltham, MA, USA). Fluorescence images were obtained using a Leica STELLARIS 5 confocal microscope (Leica, Wetzlar, Germany) and then analyzed using Leica confocal software (2.61.15) and processed in ImageJ 1.54f (LOCI, Madison, WI, USA).

### 4.8. IFA

The levels of sTREM-1 secretion were tested using a Human TREM-1 ELISA kit (ABclonal, Woburn, MA, USA) in accordance with the manufacturer’s protocols.

### 4.9. MALDI Analysis

The MALDI analysis was performed as described in [39].

### 4.10. Animal Experiments

All animal protocols used in this study were reviewed and approved by the Institutional Animal Care and Use Committee (approval number 423/25, 7 July 2025). All methods performed in this study were in accordance with the IACUC guidelines and regulations. These guidelines are equivalent to the ARRIVE guidelines and, therefore, all methods were performed in accordance with the ARRIVE guidelines (Appendix A).

Female ICR mice weighing 25 ± 1.1 g (7-week-old) were obtained from the Animal Breeding Facility of IBCh RAS (Bioresource Collection supported by the Ministry of Science and Higher Education of the Russian Federation, contract No. 075-15-2025-486). During the experiment, all mice were maintained in a specific pathogen-free (SPF) state under a strict light cycle (lights on at 07:00 h and off at 19:00 h), at 22 ± 2 °C and 50 ± 10% relative humidity. Animals were provided ad libitum access to a standard laboratory autoclavable rodent diet.

Twenty-one mice were used for the present study. The animals were randomly assigned to three groups with seven animals in each group: (1) control, intact animals; (2) intravenous injection of physiological saline (100 μL 0.9% NaCl) as a placebo during LPS-induced sepsis; (3) intravenous injection of peptide N7.1 (120 µg in 100 µL saline per mouse) during LPS-induced sepsis. LPS-induced sepsis was stimulated by an intraperitoneal injection of 0.6 mg lipopolysaccharides from *Salmonella enterica* Serotype Typhimurium (Sigma-Aldrich Co., St. Louis, MO, USA). Due to the small fluctuation in animal weight, weight-based dosing of LPS was not used. Drug treatment (saline or peptide N7.1) was carried out 1 h after LPS-induced sepsis. Survival of the mouse was observed at 2 h, 8 h, 12 h, 24 h, 48 h, 72 h, 96 h, and 120 h after LPS administration.

### 4.11. Statistical Analysis

The data are presented as means ± standard deviations. Each experiment was replicated at a minimum of three times. Data were analyzed using Statistica 6.1 (StatSoft^®^, Tulsa, OK, USA) software using Student’s *t*-tests for experiments on cell treatments with a single agent and a two-way ANOVA for experiments on cell treatments with two or more agents (see individual figure legends). The results are presented as average values ± SDs. The value of *p* < 0.05 was considered statistically significant. Data visualization was accomplished using GraphPad Prism 6 [40].

## 5. Conclusions

In this study, we have found two epitopes of the Hsp70 protein, capable of inducing antitumor cytotoxic lymphocytes in TREM-1 dependent and TREM-1 independent pathways. The shortened N7.1 peptide from one of these epitopes is able to prevent interaction of the TREM-1 receptor with its ligands in vitro and animal death from LPS-induced sepsis in vivo. The results obtained can be used in antitumor and anti-inflammation therapy.

## Figures and Tables

**Figure 1 ijms-26-09750-f001:**
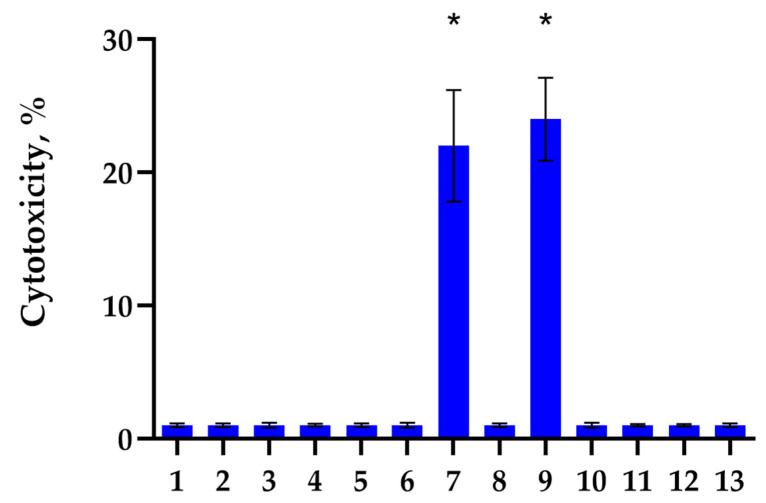
Cytotoxic activity of PBMC incubated for 6 days with peptide fractions of Hsp70 after limited trypsinolysis and separation on a Superdex-Peptide column via HPLC. The peptide fractions were added to PBMC for 6 days, and on day 6, activated lymphocytes were transferred to K562 tumor cells for 24 h to test their cytotoxic activity. (* *p*-value < 0.05).

**Figure 2 ijms-26-09750-f002:**
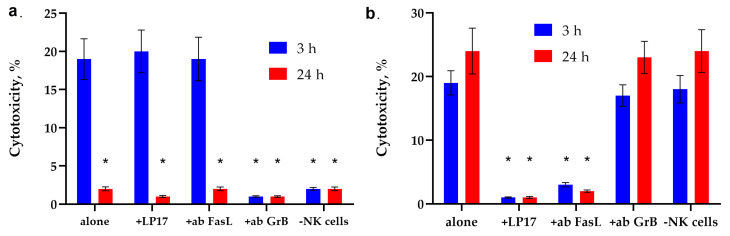
The cytotoxic activity of PBMC activated by peptides of fractions No. 9 and No. 7. (**a**) Cytotoxic activity of the PBMC activated by peptide fraction No.9 after 3 or 24 h of incubation with K562 cells in the presence of the blocking peptide LP17, an antibody to FasL (1:100, 24 h) and after removal of NK cells via magnetic separation. (**b**) Cytotoxic activity of the PBMC activated by peptide fraction No.7 after 3 or 24 h of incubation with K562 cells in the presence of the blocking peptide LP17, an antibody to FasL (1:100, 24 h) and after removal of NK cells via magnetic separation. *n* = 5 for each point (* *p*-value < 0.05, *t*-test vs. alone).

**Figure 3 ijms-26-09750-f003:**
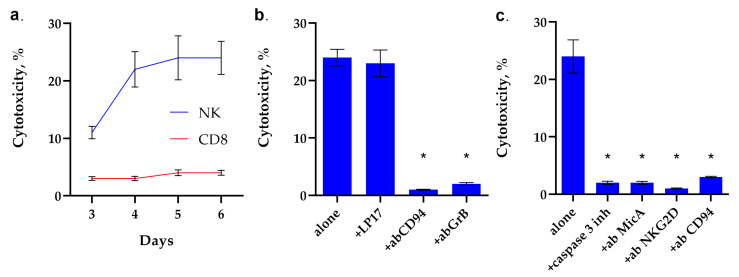
Analysis of the cytotoxic activity of PBMC activated by TKD peptide. (**a**) Cytotoxic activity of the PBMC CD8+ and NK fractions purified via magnetic separation after 3–6 day-long incubation with TKD peptide. (**b**) Cytotoxic activity of the PBMC activated by TKD peptide for 6 days after 24 h of incubation with K562 cells in the presence of the blocking peptide LP17, an antibody to CD94 (1:100, 24 h) and an antibody to Granzyme B (1:100, 24 h). (**c**) Cytotoxic activity of the PBMC activated by TKD peptide for 6 days, 24 h of incubation with K562 cells in the presence of the caspase 3 inhibitor, an antibody to MicA (1:100, 24 h) and NKG2D (1:100, 24 h). *n* = 5 for each point (* *p*-value < 0.05).

**Figure 4 ijms-26-09750-f004:**
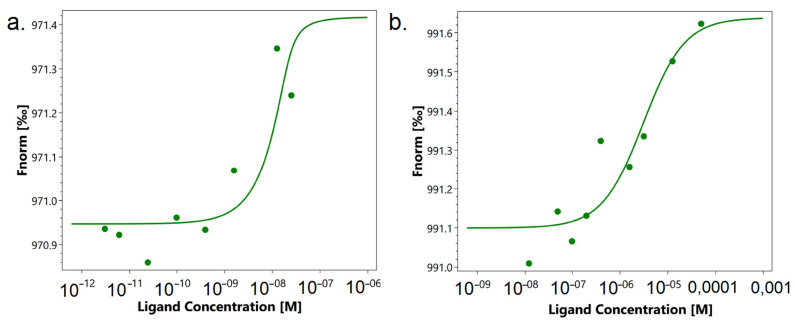
Microscale thermophoresis data for the interaction of N7 peptide with TREM-1 (**a**) and TKD peptide with TREM-1 (**b**). Each experiment was carried out in triplicate, and the most common data are shown.

**Figure 5 ijms-26-09750-f005:**
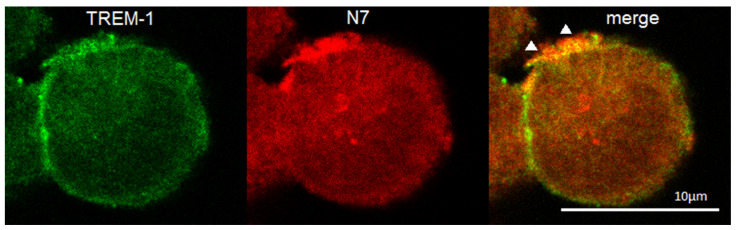
Confocal micrograph of the THP-1 cell stained by anti-TREM-1 (green), anti-Hsp70 (red) and layers superposition (yellow). Arrows indicate colocalization region. About 62% of all cells were double-stained. *n* = 4. Dimension is shown on the picture.

**Figure 6 ijms-26-09750-f006:**
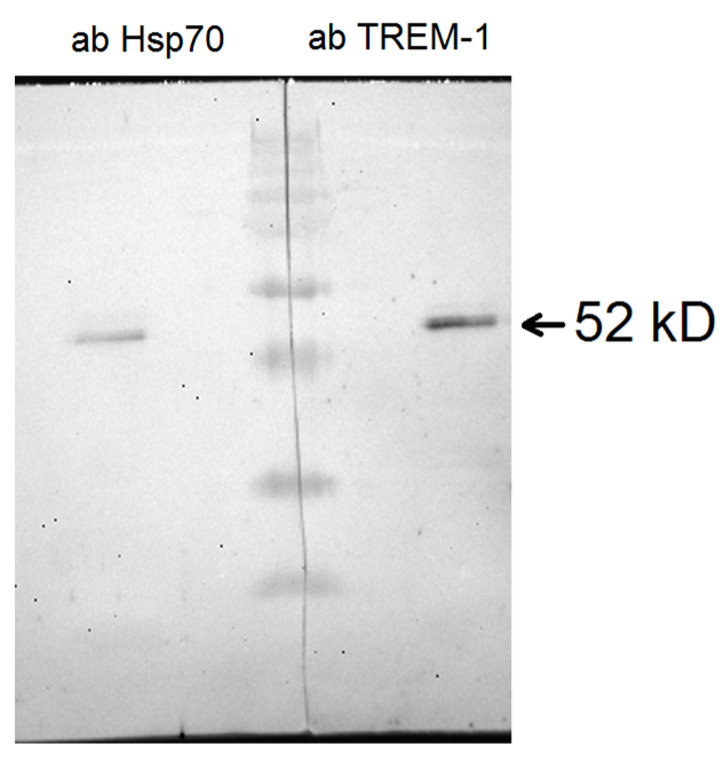
Crosslinking N7 peptide with TREM-1 on the monocyte surface in the presence of BS^3^. Cells were lysed and purified using Dynabeads conjugated with anti-TREM-1 antibodies. This material was resolved by 10% SDS-PAGE followed by Western blotting and detected with rabbit anti-Hsp70 antibody (left) and mouse anti-TREM-1 antibody (right).

**Figure 7 ijms-26-09750-f007:**
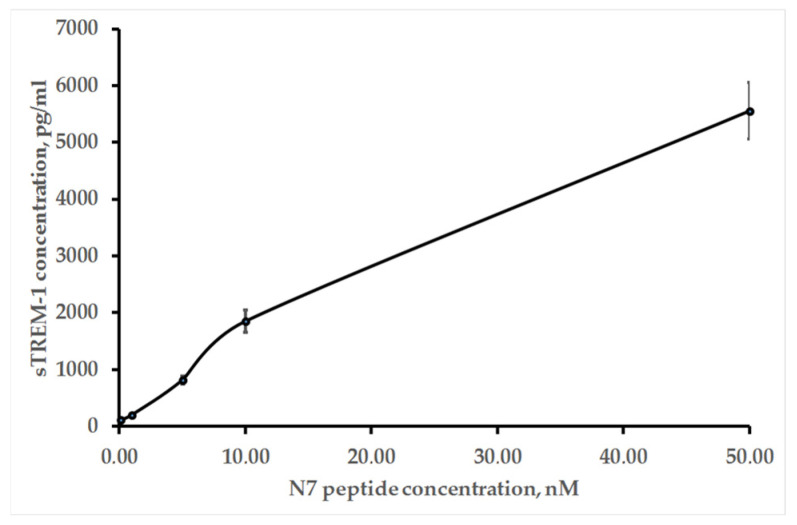
ELISA of sTREM-1 in a conditioned medium of monocytes incubated with the various concentrations of the N7 peptide for 20 h.

**Figure 8 ijms-26-09750-f008:**
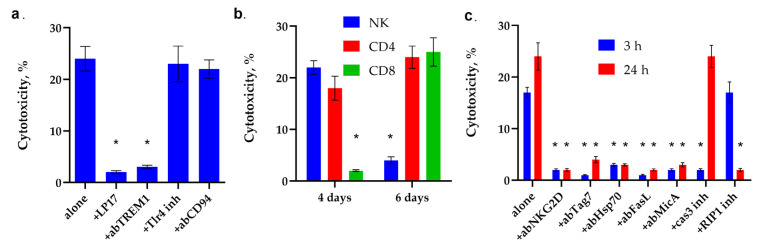
Analysis of the cytotoxic activity of PBMC activated by N7 peptide. (**a**) Cytotoxic activity of the PBMC activated by N7 peptide for 6 days after 24 h of incubation with K562 cells in the presence of the blocking peptide LP17, an antibody to TREM-1 (1:100, 24 h), an antibody to CD94 (1:100, 24 h) and an inhibitor of TLR4 (* *p*-value < 0.05). (**b**) Cytotoxic activity of the PBMC CD4+, CD8+ and NK fractions purified via magnetic separation after 4 or 6 day-long incubation with N7 peptide. (* *p*-value < 0.05, *t*-test vs. 0 day) (**c**) Cytotoxic activity of the PBMC activated by N7 peptide for 6 days after 3 or 24 h incubation with K562 cells in the presence of an antibody to MicA (1:100), NKG2D (1:100), Tag7(1:100), Hsp70 (1:100), FasL(1:100), caspase 3 inhibitor or RIP1 kinase inhibitor. *n* = 5 for each point (* *p*-value < 0.05, *t*-test vs. alone).

**Figure 9 ijms-26-09750-f009:**
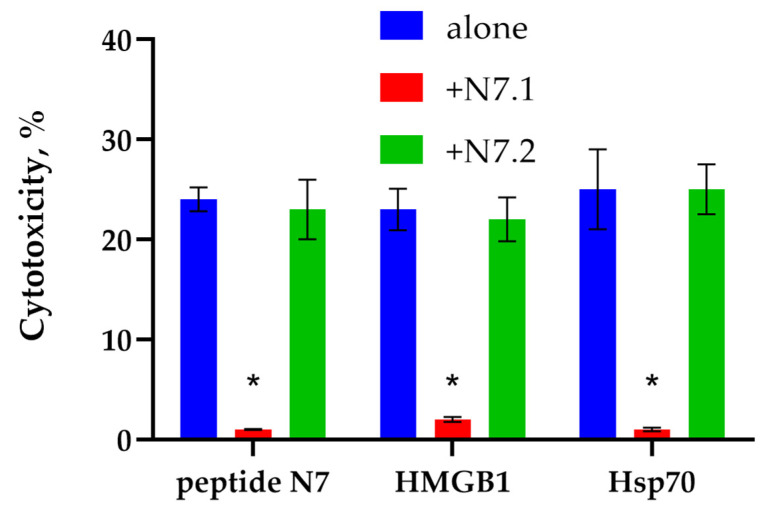
Cytotoxic activity of the PBMC activated by N7 peptide, HMGB1 or Hsp70 for 6 days after 24 h of incubation with K562 cells in the presence of the H7.1 or H7.2 peptides. point (* *p*-value < 0.05).

**Figure 10 ijms-26-09750-f010:**
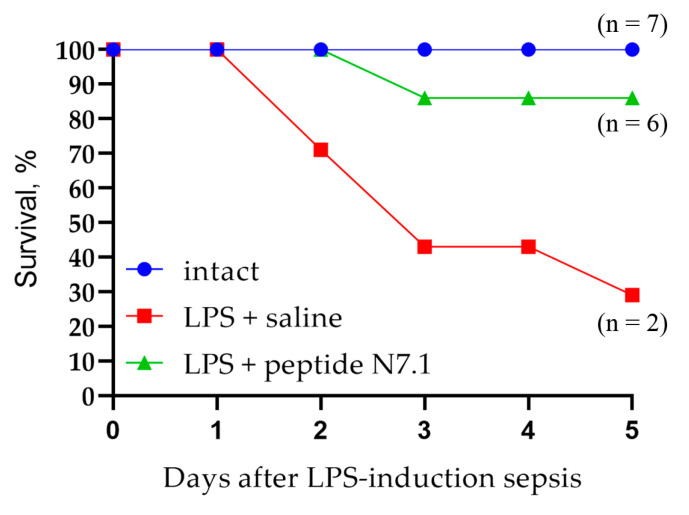
Survival of ICR mice with LPS-induced sepsis treated with saline and N7.1 peptide.

## Data Availability

The original contributions presented in this study are included in the article. Further inquiries can be directed to the corresponding author.

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
