# Peer review of "Hsp70 Peptides Induce TREM-1-Dependent and TREM-1-Independent Activation of Cytotoxic Lymphocytes"

_ijms, 2025, doi:10.3390/ijms26199750_

Round 1
Reviewer 1 Report
Comments and Suggestions for Authors
Yurkina, et al. described the biological activity of Hsp70-derived peptides to stimulate cytotoxic signatures in various lymphocyte subsets of PBMC cells, through TREM1 dependent and independent mechanism. The data indicate that two specific peptides obtained via limited enzymatic digestion of recombinant Hsp70 carry capacity to trigger MHC-independent cytotoxic response in PBMC – one (identified as N9 fraction) causing short-term cytotoxic response dependent on Granzyme B secretion and the presence on NK cells, the other (referred to as N7 fraction) leading to a more sustainable over 24hr cytotoxicity of PBMC fraction requiring Fas-FasL signaling and direct engagement of TREM1 receptor.
The manuscript presents a range of convincing data combinedly collected from biochemical binding assays, in vitro tissue culture, and live animal sepsis model systems to corroborate the findings, that are both sufficiently original and novel – particularly in part of defining longitudinal dynamic changes in specific lymphocyte subsets of PBMC fraction that upregulate their cytotoxic activity in response to N7 peptide stimulation – to anticipate its interest for broad reader audience and justify its publication in It. J. Mol. Sci.
However, the following aspects need to be clarified or elaborated on in the originally submitted manuscript:
- Several biological concepts or prior information critical to understand the data are either only fleetingly mentioned or fully ignored in introductory part (or in corresponding results sections). For example, including a paragraph in Introduction part outlining the known role of TKD peptide as a potentiator of NK activity will be essential to appreciate the findings on the activation properties of N9 fraction that contains peptide largely overlapping TKD sequence. The same notion also applies to LP17 peptide as inhibitor of TREM1 signaling. Furthermore, the role of TREM1 pathway in facilitating MHC-independent activation of myeloid (and indirectly cytotoxic lymphoid) cells deserves a special mentioning to appreciate the choice of the K562-bases in vitro cytotoxicity assay to evaluate cytotoxic responses of PBMC fractions activated by Hsp70-derived peptides. Authors should expand the introductory section to elaborate on these essential aspects of TREM1 biology to prepare readers for better understanding of reported findings.
- Several pieces of data, figures, and corresponding parts in Materials and Methods section need additional essential information. For example, in Fig. 2 it is difficult to understand the exact experimental flow – for how long were PBMC cells incubated with N7 and N9 peptide fractions prior (or simultaneously?) to incubation with K562 cells, and at what time point were blocking antibodies or inhibitory peptide LP17 added?
- 5 interpretation is not clear and in fact rather misleading. The text claims that evident co-localization of signals obtained by decorating THP-1 cells with anti-TREM1 and anti-Hsp70 antibodies serves the proof of interaction between membrane TREM1 and exogenously added N7 peptide. However, THP-1 cells actually express Hsp70 endogenously and used in assay polyclonal anti-Hsp70 antibody will most certainly bind other partes of Hsp70- polypeptide, not just N7 epitope and thus will stain endogenously produced Hsp70 that – e.g. in its extracellular secreted form – can also co-localize with TREM1. To the least, authors should include an additional panel demonstrating THP-1 cells not subjected to N7 treatment (or treated with some unrelated mock peptide of similar length and/or amino acid composition) and double stained with anti-TREM1/anti-Hsp70 antibodies to verify the dependence of colocalization signal from N7 peptide.
- Regarding the same Fig. 5 – the reported shift from cytotoxic response at day 4 mediated in large part by innate lymphocyte (NK) fraction of PBMC to predominantly adaptive lymphocytes-based cytotoxicity at day 6 is clearly an intriguing observation. Authors should try to demonstrate if this switch is related to the availability and/or dynamic change of TREM1 receptor expression itself by evaluating (e.g. by Western blot analysis) TREM1 expression on three different PBMC fractions at Day 4 and Day 6 after exposure of PBMC to N7 peptide.
- Describing perhaps the most critical finding of the work in Fig. 8 (activation time-dependence of cytotoxic response in CD4+ vs. CD8+ vs. NK PBMC fractions), authors should include more experimental information e.g. about absolute counts of different lymphocyte types isolated from PBMC at 4 days and 6 days after N7 peptide activation and the number of lymphocytes in sub-populations analyzed for cytotoxicity at different time points post-induction.
- Sepsis shock induction experiment, while outlining a possible avenue of therapeutic application for study findings, should also be more clearly described in its experimental design part. Were animals pre-treated with N7.1 peptide prior to LPS challenge, or has the N7.1 peptide been co-administered with an LPS dose, or at some time point upon LPS treatment? Since mice in LPS/saline and LPS/N7.1 treatment groups appear to receive the same mass amount of LPS (0.6mg), have the LPS treated animal been randomized by body weight in saline vs. N7.1 treatment arms? As a minor remark, including the experimental power (n=7/study arm) on the graph will help readers to appreciate the survival percentage readouts in terms of animal numbers.
- While the manuscript is written in a good quality, easy to understand and follow language, authors are encouraged to review the text one more time for obvious textual oversights. For example, the ultimate paragraph of Introduction section (page 2 of 16) states “The main tasks were: 1). establishment of the…”, but never mentions what items 2), 3), etc. actually are. On page 8 of 16, authors state that “In our experiments, as previously described, NK cells are also killed by the interaction of NKG2D receptor with MicA…” which does not make a lot of sense as these are in fact NK cells that do killing of tumor cells by NKG2D-MicA mediated GrB release.
- Section on Materials and Methods, while sufficiently informative and detailed, can be improved by better description of experimental timelines and spelling our several abbreviations. Also, caspase 8 inhibitor and necrostatin 1 (which is actually a blocking agent for RIPK1 kinase) are mentioned in Materials’ section 4.5, but nowhere else in the paper.

Author Response
Several biological concepts or prior information critical to understand the data are either only fleetingly mentioned or fully ignored in introductory part (or in corresponding results sections). For example, including a paragraph in Introduction part outlining the known role of TKD peptide as a potentiator of NK activity will be essential to appreciate the findings on the activation properties of N9 fraction that contains peptide largely overlapping TKD sequence. The same notion also applies to LP17 peptide as inhibitor of TREM1 signaling. Furthermore, the role of TREM1 pathway in facilitating MHC-independent activation of myeloid (and indirectly cytotoxic lymphoid) cells deserves a special mentioning to appreciate the choice of the K562-bases in vitro cytotoxicity assay to evaluate cytotoxic responses of PBMC fractions activated by Hsp70-derived peptides. Authors should expand the introductory section to elaborate on these essential aspects of TREM1 biology to prepare readers for better understanding of reported findings.
We thank the reviewer for careful reading of our manuscript and interesting remarks. We have added to the Introduction Section paragraphs explaining known functions of TKD peptide and LP17 peptide as the inhibiting peptide of TREM-1 receptor signaling. We also add K562 cell description and the statement that they are invisible for classical T cells due to classical MHC absence.
Several pieces of data, figures, and corresponding parts in Materials and Methods section need additional essential information. For example, in Fig. 2 it is difficult to understand the exact experimental flow – for how long were PBMC cells incubated with N7 and N9 peptide fractions prior (or simultaneously?) to incubation with K562 cells, and at what time point were blocking antibodies or inhibitory peptide LP17 added?
We thank the reviewer for careful reading and uncovering of our mistakes. We have added missed information about experimental conditions in the Figure legends, and Materials and Methods Section.
5 interpretation is not clear and in fact rather misleading. The text claims that evident co-localization of signals obtained by decorating THP-1 cells with anti-TREM1 and anti-Hsp70 antibodies serves the proof of interaction between membrane TREM1 and exogenously added N7 peptide. However, THP-1 cells actually express Hsp70 endogenously and used in assay polyclonal anti-Hsp70 antibody will most certainly bind other partes of Hsp70- polypeptide, not just N7 epitope and thus will stain endogenously produced Hsp70 that – e.g. in its extracellular secreted form – can also co-localize with TREM1. To the least, authors should include an additional panel demonstrating THP-1 cells not subjected to N7 treatment (or treated with some unrelated mock peptide of similar length and/or amino acid composition) and double stained with anti-TREM1/anti-Hsp70 antibodies to verify the dependence of colocalization signal from N7 peptide.
We thank the reviewer for useful information about THP-1 cells expressing endogenous Hsp70. We have added in the Supplemental Material one of numerous confocal controls demonstrating no Hsp70 detection at the experimental conditions in the cells, that do not receive N7 peptide.
Regarding the same Fig. 5 – the reported shift from cytotoxic response at day 4 mediated in large part by innate lymphocyte (NK) fraction of PBMC to predominantly adaptive lymphocytes-based cytotoxicity at day 6 is clearly an intriguing observation. Authors should try to demonstrate if this switch is related to the availability and/or dynamic change of TREM1 receptor expression itself by evaluating (e.g. by Western blot analysis) TREM1 expression on three different PBMC fractions at Day 4 and Day 6 after exposure of PBMC to N7 peptide.
Our previous findings show that the signal transduction from TREM-1 receptor to the cells subpopulations that possess cytotoxic activity is indirect. TREM-1 is activated by its ligands on the surface of monocytes. If we deplete PBMC from monocytes fraction, the remaining cells do not react on TREM-1 stimulation. Monocytes upon TREM-1 activation start to produce proinflammatory cytokines IL-1, IL-6, TNF, and these cytokines activate CD4+ subpopulation of T cells. CD4 deprived PBMC also fail to develop cytotoxic effect. And in fact the activated CD4+ T cells than produce IFNg and IL-2, that are essential for cytotoxic populations development. In more details this TREM-1 dependent lymphocytes activation is described in our previous works. (ref. 19 in the manuscript, Sharapova, Tatiana N.; Romanova, Elena A.; Ivanova, Olga K.; Yashin, Denis V.; Sashchenko, Lidia P. Hsp70 Interacts with the TREM-1 Receptor Expressed on Monocytes and Thereby Stimulates Generation of Cytotoxic Lymphocytes Active against MHC-Negative Tumor Cells. Int. J. Mol. Sci. 2021, 22, 13, 6889.)
TREM-1 receptor is almost absent on the surface of normal T lymphocytes. Different timing of NK cells and lymphocytes activation results in activity of different cells at 4 and 6 day of incubation with TREM-1 activating peptide. The peak of NK activation is achieved on day 4 and after this point they decrease in numbers and more important in ability to induce tumor cell death. It seems that the active subpopulation of NK cells is lysed by programmed cell death. The cytotoxic CD8 subpopulation is activated only at 6 day of incubation with TREM-1 activating peptide.
Describing perhaps the most critical finding of the work in Fig. 8 (activation time-dependence of cytotoxic response in CD4+ vs. CD8+ vs. NK PBMC fractions), authors should include more experimental information e.g. about absolute counts of different lymphocyte types isolated from PBMC at 4 days and 6 days after N7 peptide activation and the number of lymphocytes in sub-populations analyzed for cytotoxicity at different time points post-induction.
We have added in the Supplemental Material the typical results of cytometric analysis of PBMC and isolated by magnetic separation NK and T lymphocytes, that were added to the K562 tumor cells on day 4 and 6. We also have changed Results Section to address this comment and provide more experimental data to the manuscript.
Sepsis shock induction experiment, while outlining a possible avenue of therapeutic application for study findings, should also be more clearly described in its experimental design part. Were animals pre-treated with N7.1 peptide prior to LPS challenge, or has the N7.1 peptide been co-administered with an LPS dose, or at some time point upon LPS treatment? Since mice in LPS/saline and LPS/N7.1 treatment groups appear to receive the same mass amount of LPS (0.6mg), have the LPS treated animal been randomized by body weight in saline vs. N7.1 treatment arms? As a minor remark, including the experimental power (n=7/study arm) on the graph will help readers to appreciate the survival percentage readouts in terms of animal numbers.
The animals used in this experiment have weight 25±1.1 g. The animals were randomly assigned to three groups with 7 animals in each group: LPS-induced sepsis was stimulated by intraperitoneal injection of 0.6 mg lipopolysaccharides from Salmonella enterica serotype typhimurium. Due to the small fluctuation in animal weight, weight-based dosing of LPS not used. Drug treatment (saline or peptide N7.1) was carried out 1 hour after LPS-induced sepsis. Survival of the mouse was observed at 2 h, 8 h, 12h, 24 h, 48 h, 72 h, 96 h and 120 h after LPS administration.
We have added this information in the Materials and Methods Section and also changed the Figure 10 to make it clearer.
While the manuscript is written in a good quality, easy to understand and follow language, authors are encouraged to review the text one more time for obvious textual oversights. For example, the ultimate paragraph of Introduction section (page 2 of 16) states “The main tasks were: 1). establishment of the…”, but never mentions what items 2), 3), etc. actually are. On page 8 of 16, authors state that “In our experiments, as previously described, NK cells are also killed by the interaction of NKG2D receptor with MicA…” which does not make a lot of sense as these are in fact NK cells that do killing of tumor cells by NKG2D-MicA mediated GrB release.
We thank the reviewer for careful reading of our manuscript and uncovering of our mistakes. We have changed the Introduction and Results Section to correct this issue.
Section on Materials and Methods, while sufficiently informative and detailed, can be improved by better description of experimental timelines and spelling our several abbreviations. Also, caspase 8 inhibitor and necrostatin 1 (which is actually a blocking agent for RIPK1 kinase) are mentioned in Materials’ section 4.5, but nowhere else in the paper.
We have changed the Materials and Methods Section to correct this issue.
Reviewer 2 Report
Comments and Suggestions for Authors
This manuscript describes the identification of Hsp70-derived peptides and highlights their distinct roles in TREM-1-dependent and TREM-1-independent immune responses, which is relevant for tumor immunology as well as sepsis models. Overall the study is interesting and the abstract is clear. I only have two small suggestions regarding the figures:
-
In Figure 2 and especially Figure 8b,c, please indicate more clearly which two groups were compared to obtain the reported statistical differences. This will help readers follow the results.
-
In the legend of Figure 8, the second “b” should be corrected to “c” so that the legend matches the panels.
These are minor issues, but fixing them would improve the clarity of presentation.
Comments on the Quality of English LanguageThe quality of the English language is generally good.
Author Response
This manuscript describes the identification of Hsp70-derived peptides and highlights their distinct roles in TREM-1-dependent and TREM-1-independent immune responses, which is relevant for tumor immunology as well as sepsis models. Overall the study is interesting and the abstract is clear. I only have two small suggestions regarding the figures:
In Figure 2 and especially Figure 8b,c, please indicate more clearly which two groups were compared to obtain the reported statistical differences. This will help readers follow the results.
We thank the reviewer for reading of our work and useful comments. We have changed legends to Figures 2 and 8 according to this comment.
In the legend of Figure 8, the second “b” should be corrected to “c” so that the legend matches the panels.
We are thankful to the reviewer for remarking our mistype. We have corrected this.
These are minor issues, but fixing them would improve the clarity of presentation.
Reviewer 3 Report
Comments and Suggestions for Authors
In the study, Yurina & Yashin et al., the authors have shown that the Hsp70-derived epitopes responsible for TREM-1-dependent and independent cytotoxicity have been characterized through specific peptides. The N9 peptide (AMTKDNNLLGR), largely overlapping with the known TKD sequence, activates CD94⁺ NK cells to kill tumor cells via granzyme release and apoptosis, without interacting with the TREM-1 receptor indicating TREM-1-independent cytotoxicity. In contrast, the novel N7 peptide (SDNQPGVLIQVYEGEK) that authors discovered binds to TREM-1 and activates both NK cells and CD8⁺ T lymphocytes, leading to apoptosis and necroptosis in HLA-negative tumor cells through FasL–Fas signaling, representing TREM-1-dependent cytotoxicity. A shorter version, N7.1 (SDNQPGVL), inhibits TREM-1–ligand interactions and has demonstrated protective effects in sepsis models.
Key points:
- Fraction No. 7 contains a unique 16-amino acid peptide (N7: SDNQPGVLIQVYEGEK) from the peptide-binding domain of Hsp70, with previously undescribed effects on lymphocyte cytotoxicity. In contrast, the N9 peptide shares 8 of its 11 amino acids with the well-known TKD peptide, indicating structural similarity.
- TKD activated NK cells trigger caspase-dependent apoptosis, which is blocked by a caspase-3 inhibitor. Blocking MicA on K562 cells or CD94/NKG2D on NK cells also prevents cytotoxicity, confirming that TKD activates NK cells via the CD94/NKG2D receptor complex, not through TREM-1.
- Peptide N7 binds to the TREM-1 receptor both in solution and on the cell surface shown by microscopy. A strong concentration-dependent binding signal was observed, with a dissociation constant (KD) of 1.6 nM indicating high affinity and a stable N7 TREM-1 interaction. Treating cells with the N7 peptide and a crosslinker showed that the N7 TREM-1 complex forms a dimer with a molecular weight of 52 kDa, confirmed by gel and antibody analysis.
- ELISA results show that monocytes incubated with the N7 peptide produce sTREM-1 in a dose-dependent manner, confirming that N7 interacts with and activates TREM-1. Furthermore, a 6-day incubation of PBMCs with N7 induces TREM-1-dependent cytotoxic lymphocytes that kill MHC-negative tumor cells.
- The key findings from the mice study show that in a sepsis model, mice treated with the Hsp70 N7 peptide had significantly improved survival 86% survived compared to only 29% in the untreated group demonstrating the peptide’s strong protective effect against septic shock.
The N7 peptide discovered by the authors can be considered a new activator of cytotoxic lymphocytes against MHC-negative tumor cells and can be potentially used in the development of drugs for antitumor therapy.
Minor comments:
- Abstract: An abstract is not clearly defined and would need to be rewritten. Usually, abstract consist of brief background/Introduction followed by explaining the broader scientific problem, experiments performed, key results and research findings followed by conclusions and implications. The authors have summarized the key findings but still missing the broader scientific problem which they are trying to answer is not mentioned clearly in the abstract. The authors should state that specific Hsp70-derived peptide epitopes are responsible for inducing TREM-1-dependent and TREM-1-independent cytotoxic immune responses, and how do these epitopes modulate immune cell activation and tumor cell death is unknown, so here in this study…….
- A brief introductory explanation of limited trypsinolysis (with reference) would be helpful for the readers to understand the context and the experimental approach used for identifying Hsp70 epitopes. A more detailed cell culture conditions of PBMC including the cytokines or growth factors used for maintaining the cells in the culture conditions for 6 days.
Author Response
Minor comments:
- Abstract: An abstract is not clearly defined and would need to be rewritten. Usually, abstract consist of brief background/Introduction followed by explaining the broader scientific problem, experiments performed, key results and research findings followed by conclusions and implications. The authors have summarized the key findings but still missing the broader scientific problem which they are trying to answer is not mentioned clearly in the abstract. The authors should state that specific Hsp70-derived peptide epitopes are responsible for inducing TREM-1-dependent and TREM-1-independent cytotoxic immune responses, and how do these epitopes modulate immune cell activation and tumor cell death is unknown, so here in this study…….
We thank the revierer for careful reading of our work and useful comments. We have changed the Abstract Section to address this comment.
- A brief introductory explanation of limited trypsinolysis (with reference) would be helpful for the readers to understand the context and the experimental approach used for identifying Hsp70 epitopes. A more detailed cell culture conditions of PBMC including the cytokines or growth factors used for maintaining the cells in the culture conditions for 6 days.
We have added text with the reference prior to the trypsinolysis in the Results Section for more information about used methodology.
We apologize for mistyping, the PBMC cells are cultured in mL in RPMI-1640 with 10% fetal calf serum FCS(Cytiva Livescience™, Marlborough, MA, USA). We have changed Materials and Methods Section to address this comment.